# Pupillometry reveals perceptual differences that are tightly linked to autistic traits in typical adults

**Marco Turi[1,2], David Charles Burr[1,3,4]\*, Paola Binda[1,5]\***

[1]Department of Translational Research on New Technologies in Medicine and Surgery, University of Pisa, Pisa, Italy; [2]Fondazione Stella Maris Mediterraneo, Potenza, Italy; [3]Department of Neuroscience, Psychology, Pharmacology and Child Health, University of Florence, Firenze, Italy; [4]School of Psychology, University of Sydney, Sydney, Australia; [5]CNR Neuroscience Institute, Pisa, Italy

**Abstract** The pupil is primarily regulated by prevailing light levels but is also modulated by perceptual and attentional factors. We measured pupil-size in typical adult humans viewing a bistable-rotating cylinder, constructed so the luminance of the front surface changes with perceived direction of rotation. In some participants, pupil diameter oscillated in phase with the ambiguous perception, more dilated when the black surface was in front. Importantly, the magnitude of oscillation predicts autistic traits of participants, assessed by the Autism-Spectrum Quotient AQ. Further experiments suggest that these results are driven by differences in perceptual styles: high AQ participants focus on the front surface of the rotating cylinder, while those with low AQ distribute attention to both surfaces in a more global, holistic style. This is the first evidence that pupillometry reliably tracks inter-individual differences in perceptual styles; it does so quickly and objectively, without interfering with spontaneous perceptual strategies.
DOI: https://doi.org/10.7554/eLife.32399.001

**\*For correspondence:**
dave@in.cnr.it (DCB);
paola1binda@gmail.com (PB)

**Competing interests:** The authors declare that no competing interests exist.

## Introduction

A visual scene can be perceived at various hierarchical levels of structure, from the most local elements to the global organization. A dense patch of trees is perceived as a forest at a global level, while a progressively more local analysis reveals the individual trees, then their leaves, bark, etc. The basis of global perception is the structuring into larger units the 'bits and pieces' of visual information in order to perceive objects and their relations (*de-Wit and Wagemans, 2015*). The ability to form a whole from parts, ignoring details to form meaningful classes, is a major developmental step in childhood and a component of how we define intelligence – one of the tests for IQ in the standard Wechsler Scale. Different authors have developed relatively simple perceptual tasks that have become established indexes of the preference for local or global: the block design task (*Kohs, 1920*; *Wechsler, 1955*), the Rod-and-Frame Test (*Witkin and Asch, 1948*); the Embedded Figures Test (EFT: 5); and Navon's hierarchical letters (*Navon, 1977*). The preference for local or global is also part of a general way of feeling and behaving, as described by personality traits. Perhaps, the clearest example is the tendency for local or detail-oriented perception in Autistic Spectrum Disorders (*Happé and Frith, 2006*; *Mottron et al., 2006*; *Plaisted et al., 1999*), compared with the more global or holistic perception in typical adults (*Navon, 1977*): autistic observers show slower responses to global structure (*Van der Hallen et al., 2015*) and enhanced processing of local features (*Mottron et al., 2006*; *Muth et al., 2014*). There is increasing interest in determining whether differences in the preference for local or global styles are also associated with autistic traits in the typical population, supporting the dimensional view of autistic disorders, where people with and

**eLife digest** The pupils control how much light reaches the eye. They become smaller in bright light and larger in darkness to let more light in. Other factors can also affect pupil size. For example, the pupils slightly constrict when a person focuses on brighter objects and they enlarge when focusing on a darker object.

Tracking changes in pupil size can tell scientists what someone is focusing on. This can be helpful because different people distribute their attention differently. Some tend to focus on the big picture, others tune into individual details. People with autism spectrum disorders (ASD) tend to focus on the details.

Now, Turi et al. showed that measuring pupil size changes during a simple visual task could identify typical people who have milder versions of the characteristics seen in people with ASD. In the experiments, 50 young adults without a diagnosis of an ASD filled out a questionnaire designed to assess how many ASD-like traits they have. Next, the participants watched an illusion of a cylinder with a light and dark side rotating. As they watched, the pupil size of people with more ASD-linked behaviors fluctuated more than the pupil size of those with few such characteristics.

The pupils of people with ASD-type traits became larger when they perceived the dark side of the cylinder to be forward, and smaller when the light side appeared. This suggests they are focusing on the front of the cylinder. Future studies are needed to see if similar pupil fluctuations occur in people diagnosed with ASD. Turi et al. predict that pupil changes will be even more dramatic in people with ASD. If this is the case, these pupil measurements could be used to help diagnose ASD or determine the severity of symptoms.

DOI: https://doi.org/10.7554/eLife.32399.002

without ASD diagnosis lay along a continuum and differ only quantitatively, not qualitatively (*Baron-Cohen et al., 2001*; *Constantino and Todd, 2003*; *Ruzich et al., 2015*; *Skuse et al., 2009*; *Wheelwright et al., 2010*). However, not all attempts have been successful at showing a local-global difference in neurotypical individuals (*Cribb et al., 2016*), which may a result from the limited reliability and validity of the tests, which correlate little with each other and probably measure very different combinations of abilities and constructs (*Milne and Szczerbinski, 2009*).

Here, we report on a pupillometry-based paradigm that yields an objective index of local-global processing. We tested 50 randomly selected neuro-typical adults with variable degrees of autistic traits, as measured by the Autism-Spectrum Quotient (AQ), a tool developed to characterize the normal spectrum of subclinical autistic behaviours in the general population (*Baron-Cohen et al., 2001*). Our psychophysical task – simply reporting the apparent direction of rotation of a bistable stimulus – can be performed equally well with both local and global perceptual styles: either by attending to only the front surface, or by attending to the whole global rotation. By tagging the front and back surfaces with different luminance levels, the two styles should have different modulatory effects on pupil-size, which is has shown to be modulated by cognitive and top-down factors, such as perceived rather than physical luminance, and shifts in attention from dark to bright surfaces (*Binda and Murray, 2015*; *Binda et al., 2014*). Specifically, attending to a bright or dark surface is sufficient to evoke pupil constrictions or dilations respectively, tracking the focus of 'feature-based' attention (*Binda et al., 2014*). We therefore hypothesized that pupil-modulations would depend on whether participants attend locally to the front surface, or globally to both – implying that pupil-modulations in this task effectively index differences in the local-global preference across our participants.

## Results

We tracked pupil-size while participants viewed an ambiguous dynamic stimulus comprising 150 leftward-moving white dots and 150 rightward-moving black dots, giving the immediate impression of a cylinder rotating in depth at 10 revolutions per minute (see *Figure 1A* and *Video 1*). As the depth of the dots was ambiguous, either dot-group (tagged by luminance and direction) could be seen in front, resulting in either clockwise or counter-clockwise rotation. Subjects continuously reported the perceived direction of rotation by keypress or joystick for three sessions, each lasting 10 min. The 50 participants were recruited in two groups of 25, and the two sessions were intended as self-

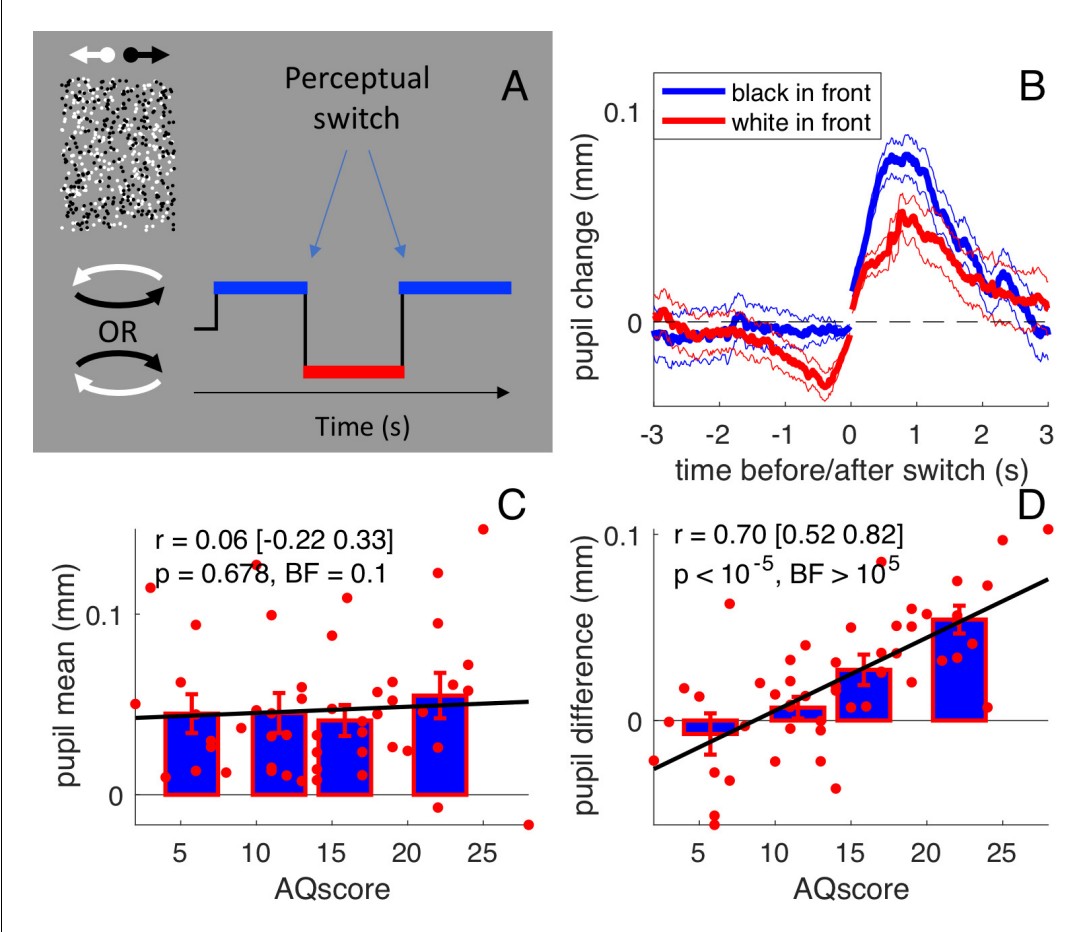

**Figure 1.** Stimulus and results of the main experiment: free viewing of the bistable cylinder. (A) Participants (N = 50) viewed two overlapping fields of dots, which created the illusion of a cylinder rotating in depth with ambiguous rotation direction. Perception oscillated between black or white dots in front: there were two types of perceptual phases. (B) Pupil size recordings were aligned to the switch of each type of perceptual phase (corresponding to zero time), from which the average pupil size in the 150 ms immediately following or preceding the switch was subtracted. Pupils dilated after the switch and constricted before it, to a different extent depending on whether perception switched to anticlockwise (implying that the black dots in front) or clockwise (white dots in front). Thick lines give averages and thin lines show 1 s.e.m. across subjects. (C) Pupil dilation after the switch ('general pupil dilation' averaged across both types of perceptual phases, over the 0:1 s interval) does not correlate with AQ scores. Text insets report Pearson's Rho values, the 95% confidence interval, and associated p-values and Bayes Factor (conventionally, the latter value indicates strong evidence in favor of the null hypothesis when smaller than 0.3, and strong evidence in favor of the alternative hypothesis when larger than 3). Bars show averages and s.e.m. for the quartiles of the AQ scores distribution and thick lines show the linear fit through the data. (D) The difference between pupil traces during the two types of perceptual phases (averaged over the intervals −1:0 and 0:1 s), that is the 'luminance-dependent' pupil modulation, correlates with subjects' AQ score. All conventions as in C.

DOI: https://doi.org/10.7554/eLife.32399.003

replications of the experiment (see below, Figure 3). On average, perceived direction alternated every 5.5 s (0.21 ± 0.01 perceptual switches per second); the between-subject variability of switch rate did not correlate with autistic traits (Pearson's r = 0.02 [-0.26 0.29], p=0.901, Bayes Factor [BF] =0.1).

*Figure 1B* shows the timecourse of pupil-size, synchronized to the perceptual switch and averaged over all subjects, separately for perceptual phases with black or white foreground (blue and red, respectively). Two distinct kinds of pupil modulations are apparent. First, as reported previously for binocular rivalry (*Einhäuser et al., 2008*), the pupil dilated transiently at or just after each perceptual switch; this effect is seen on both blue and red traces, meaning that the dilation occurs irrespective of whether perception switches toward a black or a white foreground. However, there is also a second kind of pupil modulation that depends on the direction of the shift: the pupil was more

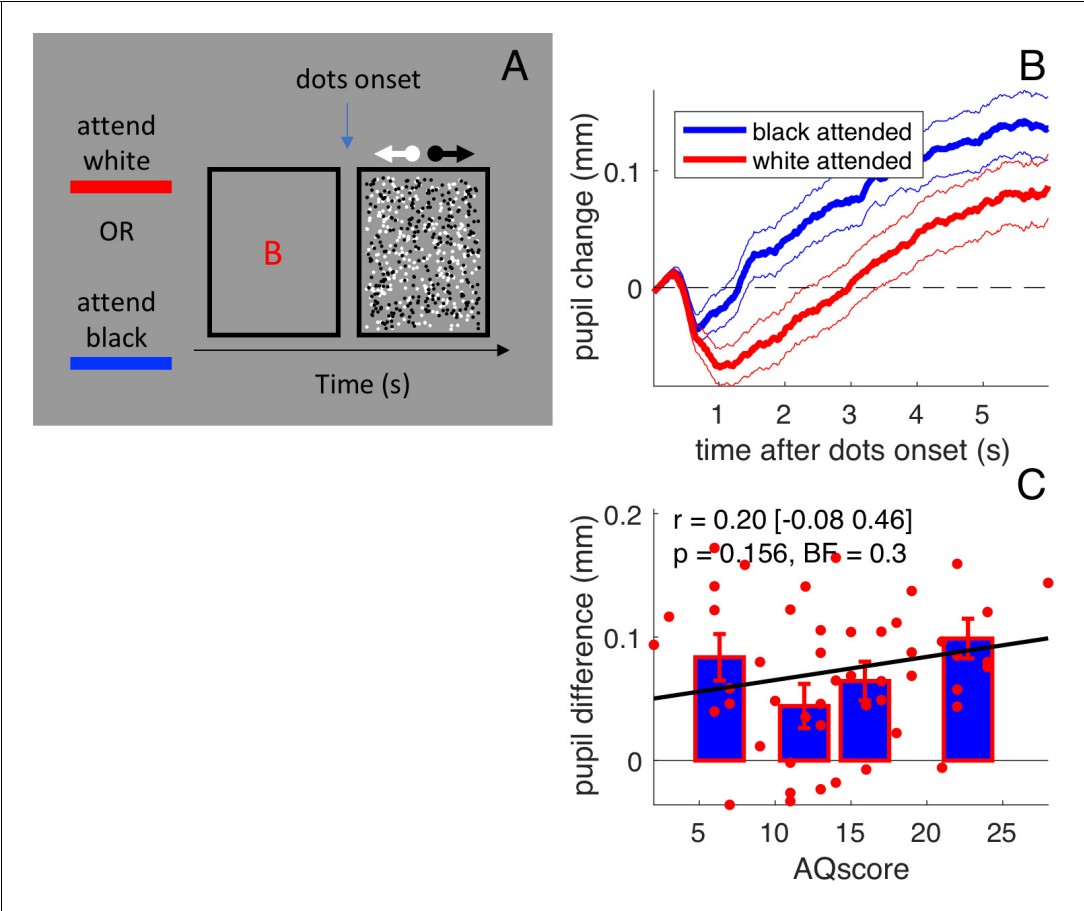

**Figure 2.** Stimulus and results of forcing attention to one surface. (A) Participants (N=50) viewed the same two overlapping fields of dots as in the main experiment, but we presented them briefly (6 sec) and preceded them with a cue letter that instructed subjects to which surface they should attend. (B) Pupil size recordings were aligned to the stimulus onset and the average pupil size in the preceding 150 ms was subtracted. Pupils dilated over the course of the stimulus, to a different extent depending on which surface was cued (more dilated when the dark surface was attended). Thick lines give averages and thin lines show 1 s.e.m. across subjects. (C) The pupil difference between trials where the black or the white surface was attended (averaged over the [1:3]s interval) does not correlate with AQ scores. All conventions as in *Figure 1*.
DOI: https://doi.org/10.7554/eLife.32399.004

dilated when the foreground was black and more constricted when it was white. This is similar to the pupillary modulation that occurs during binocular rivalry between stimuli of different luminances, leading to pupillary constriction when the brighter stimulus dominates (*Naber et al., 2011*).

This luminance-dependent modulation is a purely perceptual effect, as the stimulus luminance is constant throughout the experiment. The constriction when the foreground was white probably results from participants attending more to the foreground than the background, as it is known that attending to white constricts the pupil (*Binda et al., 2013*). On the other hand, if participants were to distribute attention evenly between the back and the front surfaces (or rapidly switch attention between them), no pupil difference would be expected. As these two strategies correspond well to the local and global perceptual styles associated with high and low autistic traits (*Happé and Frith, 2006*; *Grinter et al., 2009*), pupil modulation should be more prominent in participants with higher AQ scores. *Figure 1D* shows that, in our sample of typical young adults, there was considerable variation in luminance-dependent pupil change pupil modulation. The signed amplitude of the modulation was significantly positive in 20 out of the 50 participants. Crucially, the amplitude of modulation, which we suggest is an index of perceptual style, was highly correlated with AQ scores (r = 0.70,[0.52:0.82] $p<10^{-5}$, BF $>10^5$). Importantly, the correlation was specific to this particular pupillometric index: AQ scores did not correlate with the general dilation that followed all perceptual switches (r = 0.06,[-0.22:0.33], p=0.678, BF = 0.1: *Figure 1C*).

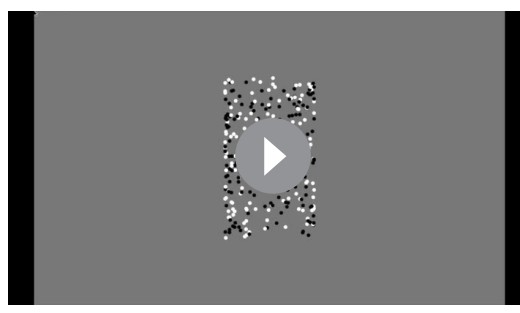

**Video 1.** the bistable cylinder
DOI: https://doi.org/10.7554/eLife.32399.005

To test the idea that the variability of pupil responses emerges from different (local-global) perceptual styles, we performed two further experiments. Firstly, we attempted to induce changes in viewing strategy by changing the instructions given to the participants, and looked for concomitant changes of pupil modulation. We retested twice more a small subset of 10 participants, under two different instruction sets: 'try to attend to both surfaces and see the cylinder rotate as a single unit'; or 'focus attention on the front surface alone'. Most subjects reported that they found the instructions difficult to follow, and that they tended to lapse into their more natural viewing style. Nevertheless, the instructions significantly affected the luminance-dependent pupil modulation (0.005 ± 0.007 mm when instruction favoured the global strategy; 0.029 ± 0.008 mm when they favoured the local strategy), which was greater when the local strategy was encouraged (paired t-test: t:(9) = −2.72, p=0.024). This supports the suggestion that pupil modulation is driven by perceptual style, and further also shows that the pupil can be a more sensitive (as well as more objective) probe of perceptual style than are subjective reports.

In a second experiment, we modified the procedure to encourage all participants to view the stimulus in the same *local* manner, attending to only one surface. Instead of the continuous presentation used before, we presented brief (6 s) bursts of the stimulus and instructed subjects to which surface they should attend (*Figure 2A*). To monitor compliance, participants reported the number of speed changes in the dots of the cued surface (ignoring changes in the other surface), which they did with an average of 62 ± 2% correct responses (uncorrelated with AQ, r = 0.24 [-0.04 0.49], p=0.090, BF = 0.5). Under these conditions, there was a strong modulation of pupil size in the averaged data, depending on whether the white or the dark surface was attended (*Figure 2B*), consistent with the reported effects of feature-based attention on pupil size (*Binda et al., 2014*). However, in this experiment, where viewing style was constrained by the task, all subjects tended to show front-color dependent pupil modulation, and it was not correlated with AQ (r = 0.20 [-0.08:0.46], p=0.156, BF = 0.3, *Figure 2C*).

Having established that the correlation between pupil difference and AQ scores is specific to the free-viewing of our bistable stimulus, we went on to check the reliability of the effect. We measured the correlation separately in the two groups of 25 participants who participated in the two sessions of the experiment (and are combined in the plots of *Figure 1B–D*). The correlation between AQ and pupil difference was strong and significant in both groups (*Figure 3A*, first subset: r = 0.75, p<10$^{-4}$, BF >10$^3$; *Figure 3B*, second subset: r = 0.64, p<0.001, BF = 54.3). We further replicated the effect in a slightly different condition, tested in 26 participants (swapping the motion directions of white and black dots, *Figure 3C*, r = 0.66 p<0.001, BF = 85.3); the three correlation coefficients are statistically indistinguishable (Fisher's Z test; all p-values>0.23), implying that the results are robust.

As a further check on the reliability of the results, we also analyzed the correlations between pupil difference in the 50-participant group (*Figure 1D*), and each of the five subscales of the AQ questionnaire (normally distributed *Figure 3D*). All but one correlation coefficients were positive and significant (*Figure 3E*). The strongest correlations were with the Communication and Social Skills subscales, and weakest with 'Attention to Detail' scores (discussed later). The bottom row of *Figure 3E* shows the correlation of each subscale with the rest of the questionnaire (the sum of the other four subscales), and also the pupil-difference index with the overall AQ score. Note that the correlation of the pupil-difference index with the total AQ score was higher than those between any subscale and the other four subscales.

The results reported in *Figures 1–3* show that pupil modulation reliably indexes attention to the front or both surfaces, depending on the participant's AQ score. In principle, perceptual style should also be measurable by psychophysical techniques, with enhanced performance on the attended surface(s) (*Carrasco, 2011*). To compare our novel pupillometry approach with more standard psychophysical methods, we measured correlations between AQ and ability to detect subtle changes on

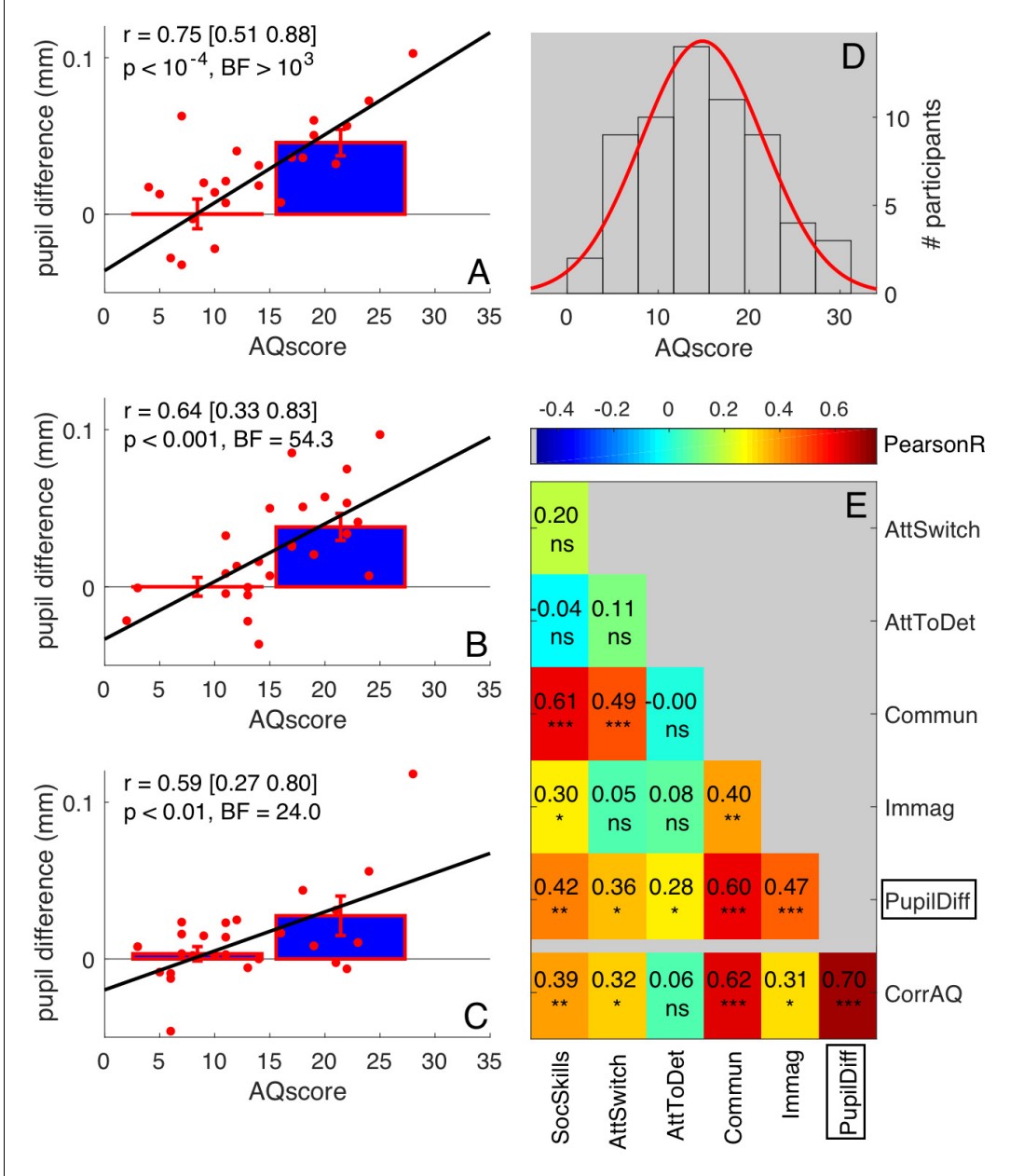

**Figure 3.** Robustness of correlations between pupil difference and AQ scores. (**A–C**) Correlations of the signed amplitude of pupil difference (or 'luminance-dependent' pupil modulation) with AQ scores separately for our three self-replications: two sessions of the main experiment tested on different subsamples of participants (A-B, 25 participants each, pooled in *Figure 1C* of the main text) and the 'swapped motion direction' experiment (C, 26 participants). Bars show averages and s.e.m. for participants with AQ score below and above the median AQ score in our pool, and continuous lines show the best fit linear function across results from all participants (red dots). Text insets give Pearson's correlation coefficients and associated p-values. (**D**) Distribution of AQ scores in our sample of participants, with best fit Gaussian function. (**E**) Correlation matrix between the pupil difference in the 'main' experiment (N = 50, pooling across panels A and B, as done in *Figure 1C* of the main text) and each AQ subscale. The bottom row shows correlations with the corrected total AQ: between the pupil difference and the total AQ, or between each subscale and the sum of the other four subscales. Pearson's r values are colour coded and given by the text with associated significance values (*p<0.05, **p<0.01, **p<0.001, ns non-significant).

DOI: https://doi.org/10.7554/eLife.32399.006

the front or rear surface. As in the main experiment, participants continuously tracked the rotation of the cylinder over 10 min long sessions; but here they also reported brief changes in speed that occurred randomly on the surface formed either by white or black dots, at the front or the rear

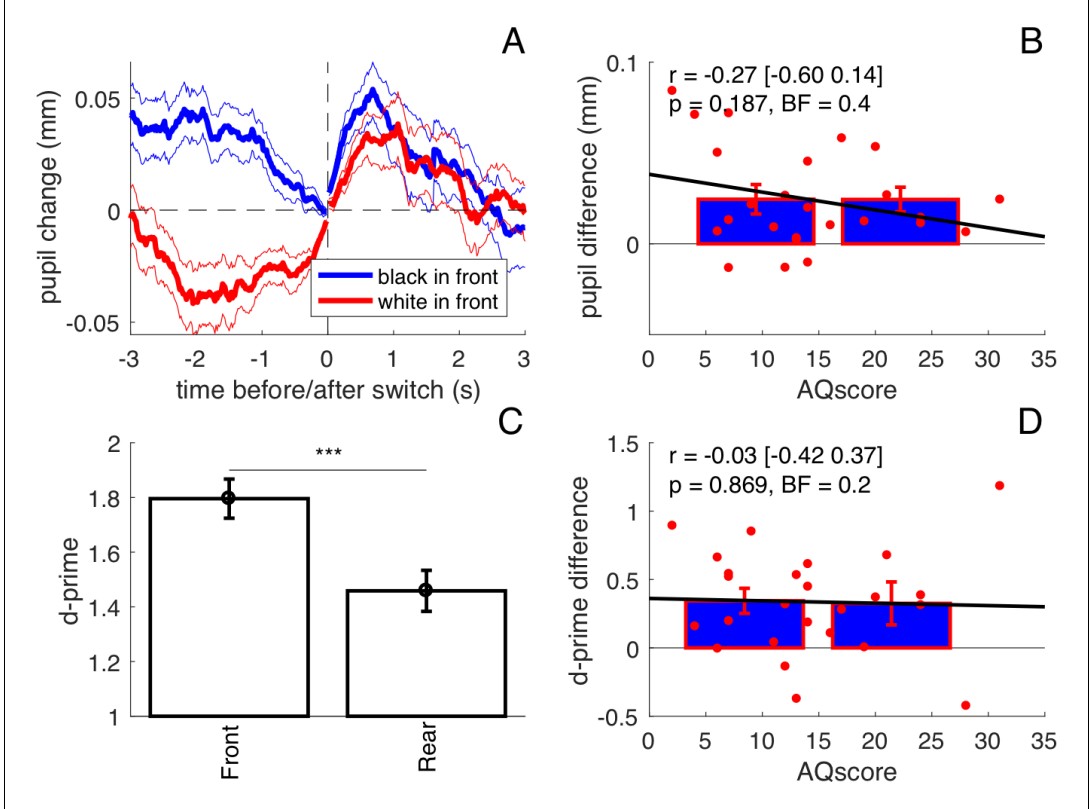

**Figure 4.** Results from the 'double task' experiment (N = 25). (**A**) Average pupil traces when the black or the white dots were seen on the foreground (same format as *Figure 1B* in the main text). (**B**) Lack of correlation between pupil difference values in the [−1:1]s interval and AQ scores. Bars show averages and s.e.m. for participants with AQ score below and above the median AQ score in our pool, and continuous lines show the best fit linear function across results from all participants (red dots). Text insets give Pearson's correlation coefficients, the 95% confidence interval, associated p-values and Bayes Factor. (**C**) Sensitivity to speed increments occurring in the foreground and background surface, measured by d-prime and averaged across participants; error bars show s.e.m.; stars show the results of a paired t-test comparing the two sets of d-prime values (***p<0.001). (**D**) The sensitivity difference between foreground and background does not correlate with AQ. Same conventions as in panel B.
DOI: https://doi.org/10.7554/eLife.32399.007

depending to the perceived rotation of the cylinder. As in the main experiment, pupils dilate more when the foreground is black compared with when it is white (*Figure 4A*). Speed-change detection performance was generally better when targets occurred on the front than on the rear surface (*Figure 4C* paired t-test on d-prime values: t(24)=4.43, p<0.001). However, AQ correlated neither with the difference in performance (*Figure 4D*, r = −0.03 [-0.42 0.37], p=0.869, BF = 0.2) nor with the pupil difference (*Figure 4B* r = −0.27 [-0.60 0.14], p=0.187, BF = 0.4). This shows that the double-task setting (necessary to obtain a psychophysical index of attention distribution) interfered with participants' natural viewing style, biasing all towards a 'local' processing style, focusing on one surface at a time to detect the speed changes. In most cases, the surface was the front one. This clearly undermined the possibility of estimating the inter-individual differences that spontaneously emerge in the free-viewing conditions of the main experiment.

## Discussion

We measured pupil size while subjects viewed a structure-from-motion stimulus perceived as a cylinder rotating in 3D with bistable direction. In some but not all participants, pupil-size modulated in synchrony with their bistable perception, more constricted when the white surface was perceived in front. The magnitude of the modulation was tightly correlated with the Autistic Quotient scores, consistent with the view that stronger autistic traits accompany a preference for focusing on local detail, as opposed to globally attending the whole stimulus configuration. Some participants spontaneously

reported their perceptual experience: of these some reported that they attending locally to only the front surface, while others reported that they had to attend globally to the whole configuration, both front and rear, in order to perceive the rotation. These spontaneous reports (respectively from participants with high and low AQ) provide qualitative support for our interpretation.

Although preference for local versus global has been extensively investigated and has come to be a defining feature of Autistic Spectrum Disorders (*Van der Hallen et al., 2015*; *Muth et al., 2014*), its underlying mechanisms are essentially unknown. It could be a perceptual style, depending on differences in the bottom-up transmission of sensory information (*Mottron et al., 2006*; *Mottron et al., 2003*; *Plaisted et al., 2003*), but it could also have a cognitive or attentional basis, depending on a preference to maintain a narrow focus of attention (*Happé and Frith, 2006*; *Plaisted et al., 1999*) or a relative inability to divide or switch attention between multiple targets (*Behrmann et al., 2006*; *Frith and Happé, 1994*; *Happé and Frith, 1996*; *Happé and Booth, 2008*). Our results are equally supportive of all three possibilities, as are the classic tests for global-local preference, such as Navon letters or embedded figures. While our results do not distinguish between the different possible explanations of perceptual styles, they clearly show that autistic traits are associated with a more local perceptual style.

There is an ongoing dispute about whether autistic traits are distributed continuously amongst the population with ASD diagnosis at an extreme, or whether the distinction is more categorical (*Baron-Cohen et al., 2001*; *Constantino and Todd, 2003*; *Ruzich et al., 2015*; *Skuse et al., 2009*; *Wheelwright et al., 2010*). The hypothesis of a 'Broader Autistic Phenotype' strongly predicts that local-global preference should be continuously distributed and correlated with autistic traits in the neurotypical population (*Cribb et al., 2016*). However, many of the classic tests show very low or insignificant correlations with autistic traits (*Cribb et al., 2016*). Our experiment describes a pupillometry index with the strongest correlation with autistic traits reported so far in a large group of participants, explaining about 50% of the variance in AQ scores.

This correlation may appear surprisingly high, given that it is measured between two different tasks. However, each of these separate tasks has high reliability. The test-retest reliability of the autistic questionnaire has been reported to be around 0.75 (*Baron-Cohen et al., 2001*; *Hoekstra et al., 2008*; *Kurita et al., 2005*). In the 22 participants who performed both the main experiment and the swapped-direction, the correlation of pupil modulation in the two tasks was 0.68 ([0.36:0.86], p<0.001, BF = 62.7). Considering the 95% confidence interval range (r = 0.70, [0.52:0.82]), the obtained correlations are not unrealistic, and certainly highly significant.

Although the autistic questionnaire is well validated (*Baron-Cohen et al., 2001*), not all items have the same validity. We also observed variability in the strength of correlation of the pupil index and the AQ subscales, which was strongest for the Communication and Social Skills subscales and, somewhat surprisingly, weakest for the 'Attention to Detail' scores. However, the 'Attention to Detail' subscale seems to have the lowest construct validity of the five subscales: it had the lowest performance in classifying individuals with and without ASD diagnosis in a sample of 350 adults, with 130 ASD diagnoses (*Lundqvist and Lindner, 2017*), it had the lowest correlation with the general AQ score in a sample of 2343 typical individuals (*Palmer et al., 2015*), and, in our sample, scores on the 'Attention to Detail' subscale did not correlate significantly with the overall AQ score. It is also worth noting that among the items contributing to the 'Attention to Detail' subscale, only four actually relate to perceptual phenomena (e.g. 'I usually concentrate more on the whole picture, rather than the small details."), the others being focused on memory and cognition (e.g. 'I am fascinated by numbers."). All this highlights a general difficulty in measuring local-global preference, either with self-report or with laboratory tests. However, by combining the objectivity of pupillometry and the flexibility of a perceptual task involving bistable perception, we can achieve a very reliable and precise measure of inter-individual differences.

There is growing interest in relating differences among individuals in their performance in perceptual tasks, with states or traits of their personality (*Tadin, 2015*; *Eldar et al., 2013*; *Antinori et al., 2016*; *Wilson et al., 2016*; *Maclean and Arnell, 2010*). However, the amount of explained variance is usually lower than in our experiment, which is a rare case where a physiological measure obtained from a laboratory experiment explains a large portion of inter-individual differences in the way we feel and behave.

To obtain the clear correlation with AQ, free-viewing of the bistable stimulus was essential: the correlation between pupillometry and AQ scores was eliminated when cueing attention, either

explicitly, by instructing participants to attend globally/locally or by indicating that the surface formed by white or black dots alone is task-relevant, or implicitly, by introducing a task that is best performed by attending a single surface at a time. Although previous work has shown differences in bistable viewing that are related to autism [(*Robertson et al., 2013*); but see also (*Said et al., 2013*)], we found that the dynamics of perceptual oscillations between the two directions of motion were uncorrelated with autistic traits. The correlation with AQ was only revealed only when taking pupillometry measures to index how attention was distributed during the bistable perception.

The current study strengthens previous claims that pupil size is modulated not only by light, but also by perceptual 'top-down' processes (*Binda and Murray, 2015*; *Mathôt and Van der Stigchel, 2015*; *Laeng and Endestad, 2012*). The neural substrates of these modulatory mechanisms are beginning to be unveiled, involving pre-frontal input into the circuit of the pupillary response to light (*Ebitz and Moore, 2017*), which in turn may include the visual cortex (*Binda and Gamlin, 2017*). These multiple influences converge into a simple subcortical system that controls a unidimensional variable: pupil size. As we show here, under appropriate conditions (free viewing a bistable stimulus with brighter and darker components), this variable can provide an objective physiological correlate of conscious perception, so reliable that it can successfully reveal subtle inter-individual differences that are hard to study psychophysically. The method is quick, easy, and objective, and requires only relatively simple, portable apparatus. We hope it can form the basis for tests suitable for clinical populations, possibly providing a new powerful tool for identifying anomalies in perception that are predictive of ASD.

## Materials and methods

### Subjects

We recruited a total of 62 subjects (42 women; age (mean ± SD): 25.53 ± 4.04), in three groups (25 in the first, 26 in the second and 11 in the last). All were students from the University of Pisa or Florence, in at least their third year. All reported normal or corrected-to-normal vision, and had no diagnosed neurological condition. The number of participants recruited for the study was selected to provide a large effect size as indicated by a priori power analysis (effect size: 0.50, $\alpha$ = 0.05, two-tail) that reveals that in order to reach a power (1−$\beta$) of 0.8 a sample size of 26 subjects was needed.

Experimental procedures were approved by the regional ethics committee [*Comitato Etico Pediatrico Regionale—Azienda Ospedaliero-Universitaria Meyer—Firenze* (FI)] and are in line with the declaration of Helsinki; participants gave their written informed consent.

### AQ score

All participants completed the Autistic-traits Quotient questionnaire, self-administered with the validated Italian version (*Baron-Cohen et al., 2001*; *Ruta et al., 2012*). This contains 50 items, grouped in five subscales: Attention Switching, Attention to Detail, Imagination, Communication and Social Skills. For each question, participants read a statement and selected the degree to which the statement best described them: ''strongly agree'', ''slightly agree'', ''slightly disagree'', and ''strongly disagree'' (in Italian). Items were scored in the standard manner as described in the original paper (*Baron-Cohen et al., 2001*): 1 when the participant's response was characteristic of ASD (slightly or strongly), 0 otherwise. Total scores ranged between 0 and 50, with higher scores indicating higher degrees of autistic traits. All tested subjects scored below 32, which is the threshold above which a clinical assessment is recommended (*Baron-Cohen et al., 2001*). The mean (SD) of the scores was 14.85 (6.73); scores were normally distributed (see *Figure 3D*), as measured by the Jarque-Bera goodness-of-fit test of composite normality (JB = 1.42 p=0.37).

### Apparatus

The experiment was performed in a quiet room with artificial illumination of 100 lux. Subjects sat in front of a monitor screen, subtending 41 × 30° at 57 cm distance, with their heads stabilized by chin rest. Viewing was binocular. Stimuli were generated with the PsychoPhysics Toolbox routines (*Brainard, 1997*; *Pelli, 1997*) for MATLAB (MATLAB r2010a, The MathWorks) and presented on a 22-inch CRT colour monitor (120 Hz, 800 × 600 pixels; Barco Calibrator), driven by a Macbook Pro Retina (OS X Yosemite, 10.10.5). Two-dimensional eye position and pupil diameter were monitored

either with a CRS LiveTrack system (Cambridge Research Systems) at 60 Hz, or with an Eyelink1000 Plus (SR Research) at 1000 Hz. We verified that although the two systems have different precision and accuracy, they yielded comparable results in our experiments. Both systems use an infrared camera mounted below the screen. Pupil diameter measures were transformed from pixels to millimeters after calibrating the tracker with an artificial 4 mm pupil, positioned at the approximate location of the subjects' left eye. Eye position recordings were linearized by means of a standard 9-point calibration routine performed at the beginning of each session.

## Stimuli and procedure

Different subsets of participants took part in the four experiments (main, swapped motion directions, feature-based attention, double-task). For the 'main' experiment, we recruited a total of 51 participants of which one was excluded (see below). We recruited and tested them in two groups (subjects 1–25 and 26–51), intended as self-replications each with 25 participants (after the exclusion of one participant based on criteria detailed below). Trials began with subjects fixating a red dot (0.15° diameter) shown at the centre of a grey background (12.4 cd/m$^2$). The stimulus comprised a centrally positioned 8 × 14° rectangle which appeared to be a cylinder rotating about its vertical axis (Figure 1A). The 3D illusion was generated by presenting a total of 300 randomly positioned dots (each 0.30° diameter) moving around a virtual vertical axis with an angular velocity of 60 deg/s (10 rotations per minute): the linear velocity followed a cosine function, 3.9°/s at screen centre. Dots were black (0.05 cd/m2) when they moved rightwards (half at any one time) and white (55 cd/m2) when they moved leftwards. The resulting stimulus was compatible with two perceptual interpretations: a cylinder rotating anticlockwise (when viewed from above) with black surface in the front and white surface at rear; or clockwise, with white surface in the front, black surface at rear. The two perceptual interpretations alternated spontaneously in all participants, who continuously reported their percept (clockwise or anticlockwise rotation of the cylinder), either by holding down one of two keyboard arrow keys or by joystick. There was no response button for uncertain or mixed percepts: subjects were instructed to report which of the two percepts was dominant if in doubt. The stimulus was played for 10 trials of 59 s each, during which participants continuously reported whether the rotation was clockwise or anticlockwise. Participants were instructed to minimize blinks and maintain their gaze on the fixation spot at all times, except during a 1 s inter-trial pause, marked by a change of colour of the fixation spot (which turned from red to black). Each participant completed a minimum of three runs, in a single session.

A subsample of 27 participants (19 of the first group of participants, 4 of the second group, 4 of the last group, one excluded as explained below) were also tested in the 'swapped motion direction' experiment – same as in the 'main' experiment, except that black dots moved leftward, and white dots moved rightward.

A small subset of participants (N = 10) were re-tested with the same stimuli and procedures as in the 'main experiment', but different instructions. These were meant to explicitly encourage a global or a local distribution of attention. In two separate sessions (randomized order), participants were either told to 'try to attend to both surfaces and see the cylinder rotate as a single unit' (encouraging global viewing), or to 'focus attention on the front surface alone' (encouraging local viewing). Each session lasted about 20 min and included two runs of ten trials each.

Another subsample of 25 participants (6 from the first group, 9 from the second group, 10 from the last, chosen for the disposability for a second testing session) took part in the 'double-task' experiment, with the same trial structure as the 'main' experiment. The primary task was unaltered, with the participant reporting whether they perceived clockwise or anticlockwise rotation of the cylinder (using a joystick rather than the keyboard to minimize interference with the secondary task). Meanwhile, subtle speed increments (1 frame duration, 600 deg/s angular velocity increment) occurred for either the black or the white dots (forming the front or the rear surface depending on the participant's perception), every 3 s on average (with 2 s minimum separation between speed increments). Participants were asked to press the space bar as soon as they detected a speed increment (on either surface). Any bar press within 2 s from a speed increment was counted as a hit; any bar press that happened more than 2 s away from any speed change was counted as a false alarm. D-prime values were computed from z-transformed hits and false alarms, separately for speed increments occurring on the front and rear surface. For each of these conditions, a minimum of two runs of 10 trials each were acquired (approximately 20 min).

Finally, 50 participants (18 from the first group, 22 from the second group, 10 from the last one) were tested in the 'feature-based attention' experiment. Trials were only 10 s long. During a pre-stimulus 2 s interval, no dots were shown and a letter (0.5° wide, either 'B' or 'N', for *bianco* or *nero*, Italian for white or black) was shown at fixation and cued subjects to attend selectively to only white or black dots. Next came the two groups of dots, moving with the same direction and speed as in the main experiment, but lasting only 6 s. During this time, between 0 and 3 speed increments could occur on the cued and uncued surface. Upon extinction of the dots, the participant had 2 s to report by keypress how many speed increments occurred on the cued surface, ignoring speed increments on the uncued surface. In this case the participants did not report the perceived direction of rotation of the cylinder, which may or may not have perceived as a 3D object rather than two independent clouds of dots. Participants performed well above chance, with an average d-prime of $2.36 \pm 0.09$. We tracked pupil diameter during the 6 s stimulus interval, separating trials where the white and the black dots were cued. This experiment was performed in two runs of 50 trials each (approximately 15 min).

## Analysis

An off-line analysis examined the eye-tracking output to exclude time-points with unrealistic pupil-size recordings (smaller than 1 mm, likely due to blinks, or larger than 7 mm, likely due to eyelash interference). We further excluded perceptual phases lasting less than 1 s (often finger errors) and longer than 15 s (marking trials with too few oscillations to measure bistability). These criteria led to the exclusion of one participant for the 'main' experiment, and one for the 'swapped motion direc-tion' experiment (who had less than 10 usable phases), leaving 50 participants for the 'main' experi-ment, for which the percentage of excluded phases is $27.39 \pm 2.18\%$, and 26 participants for the 'swapped motion direction' and 'double-task' experiments, for which the percentages of excluded phases were similar ($23.32 \pm 2.98\%$ and $22.58 \pm 3.21\%$ respectively). No trials and no participants were excluded for the 'feature-based attention' experiment.

In all three bistable experiments (the 'main' experiment, the 'swapped motion direction' and 'double-task' experiments), dark and white dots were equally likely seen as foreground (percentage of time of the dark foreground percept, respectively: $51.85 \pm 0.99\%$, $51.13 \pm 1.43\%$, $51.34 \pm 0.99\%$, never significantly different from 50%).

Pupil traces were parsed into epochs locked to each perceptual switch (when the subject changed reported perception). We aligned traces to the switch (zero in *Figure 1B*), and labeled the phases according to the perceived direction of rotation. For the 'feature-based attention' experiment, pupil traces were time-locked to stimulus onset and separated based on whether the black or white sur-face was cued (*Figure 2B*). For all experiments, we subtracted from each trace a baseline measure of pupil size, defined as the mean pupil size in the 150 ms immediately preceding or following the switch (for negative and positive traces, respectively), or stimulus onset for the 'feature based atten-tion' experiment. The resulting traces were averaged across trials and participants, separately for the two perceptual phases (or attention cues for the feature-based attention experiment), to give, *Figure 1B* and *Figure 2B*. From these, and also for the individual traces, we defined two sum-mary statistics: the difference of pupil traces between the two types of epochs, and overall mean pupil trace across all epochs. For the 'main', the 'swapped motion direction' and the 'double-task' experiments, these indices were computed after averaging pupil measurements over the first (or first and last) 1 s of each epoch, the minimum phase duration, ensuring that all phases contribute equally to the mean. However, we also verified our main result (correlations with AQ scores) with different epoch definitions.

For the 'feature-based attention' experiment, the difference of pupil traces between trials where the white and black dots were cued was computed in the interval between 1 and 3 s from stimulus onset (where the effect of attention is expected to peak [*Binda et al., 2014*]).

## Data availability

The datasets generated during the current study and scripts used for analyses are available on Dryad, at the following address: http://dx.doi.org/10.5061/dryad.b3ng3

## Acknowledgements

This research was supported by the European Research Council under the Seventh Framework Pro-gramme, FPT/2007–2013 (project ECSPLAIN grant agreement n. 338866), the Italian Ministry of University and Research under the projects 'Futuro in Ricerca' (grant agreement n. RBFR1332DJ) and the Fondazione Roma under the Grants for Biomedical Research: Retinitis Pigmentosa (RP) - Call for proposals 2013 - "CorticalPlasticity in Retinitis Pigmentosa: an Integrated Study from Animal Models to Humans". We thank Alessandra Viviani and Chiara Tortelli for help with data collection.

## Additional information

### Funding

| Funder | Grant reference number | Author |
| --- | --- | --- |
| Italian Ministry of University and Research | RBFR1332DJ | Marco Turi<br>David Charles Burr<br>Paola Binda |
| European Research Council | Escplain 338866 | Paola Binda |
| Fondazione Roma | Grants for Biomedical Research: Retinitis Pigmentosa (RP) - Call for proposals 2013 - "CorticalPlasticity in Retinitis Pigmentosa: an Integrated Study from Animal Models to Humans" | Paola Binda |

The funders had no role in study design, data collection and interpretation, or the decision to submit the work for publication.

### Author contributions

Marco Turi, Conceptualization, Data curation, Formal analysis, Investigation, Writing—original draft, Project administration, Writing—review and editing; David Charles Burr, Conceptualization, Supervision, Writing—original draft, Project administration, Writing—review and editing; Paola Binda, Conceptualization, Data curation, Formal analysis, Investigation, Methodology, Writing—original draft, Project administration, Writing—review and editing

### Author ORCIDs

Marco Turi ⓘD https://orcid.org/0000-0002-4495-0804
David Charles Burr ⓘD http://orcid.org/0000-0003-1541-8832
Paola Binda ⓘD http://orcid.org/0000-0002-7200-353X

### Ethics

Human subjects: Experimental procedures were approved by the regional ethics committee [Comitato Etico Pediatrico Regionale-Azienda Ospedaliero-Universitaria Meyer-Firenze (FI)] and are in line with the declaration of Helsinki; participants gave their written informed consent.

### Decision letter and Author response

Decision letter https://doi.org/10.7554/eLife.32399.012
Author response https://doi.org/10.7554/eLife.32399.013

## Additional files

### Supplementary files

• Transparent reporting form
DOI: https://doi.org/10.7554/eLife.32399.008

## Major datasets

The following dataset was generated:

| Author(s) | Year | Dataset title | Dataset URL | Database, license, and accessibility information |
|---|---|---|---|---|
| Turi M, Burr DC, Binda P | 2017 | Data from: Pupillometry reveals perceptual differences that are tightly linked to autistic traits in typical adults | http://dx.doi.org/10.5061/dryad.b3ng3 | Available at Dryad Digital Repository under a CC0 Public Domain Dedication |

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
