## [Decision Letter]

Thank you for submitting your article "Pupillometry reveals perceptual differences that are tightly linked to autistic traits in typical adults" for consideration by *eLife*. Your article has been reviewed by three peer reviewers, and the evaluation has been overseen by a Reviewing Editor and a Senior Editor. The following individuals involved in review of your submission have agreed to reveal their identity: Duje Tadin (Reviewer #1); Sebastiaan Mathôt (Reviewer #2).

The reviewers have discussed the reviews with one another and the Reviewing Editor has drafted this decision to help you prepare a revised submission.

In this manuscript, the authors use the pupil light response to track whether participants pay attention to the front of a bistable rotating cylinder (by using different brightness levels for the front and back). Crucially, they find that the extent to which participants selectively attend to the front of the stimulus (rather than to the stimulus as a whole) correlates strongly with their autism-spectrum quotient (AQ). The reviewers found the result to be both timely and interesting. They also note that while the global vs. local processing hypothesis has been longstanding in ASD work, the present approach is both elegant and objective and the control experiments are insightful.

At the same time, the reviewers noted the following concerns that need to be addressed before publication:

1) The key assumption of the paper is that switch-triggered pupil changes are indicative of individual differences in default perceptual styles. The authors support this by having additional experiments that bias all subjects toward the same perceptual style-which is a nice control. An additional strong support for this key assumption would be within-subject evidence that if a person switches her/his perceptual style that would result in a predicted modulation of switch-triggered pupil changes. Such a within-subject control would not need as many subjects, and would show that self-initiated changes in the perceptual style yield predictable pupil results-further supporting the main assumption of the paper. (This could be done with or without an objective confirmation that the perceptual style changed).

2) The relative directions of pupil changes in Figure 1 and Figure 3 are consistent with the authors' assumptions, but the sign of those changes is not. That is, in Figure 1, shouldn't going from black in front to white in front result in a decrease in pupil size? It appears that we're seeing two effects, one that is modulated by dot color and a main effect of switch that causes an increase in pupil size. One can come up with a reasonable explanation for that, but it needs to be included in the manuscript. If there are indeed two effects there, it would be interesting to see if they can be analytically separated and correlated with AQ.

3) There is concern with the strength of the correlation observed between the pupil measure and the total AQ score. The. 7 value seems extremely strong for this type of effect, and such a strong correlation between two measures is only possible if the measures themselves are highly reliable. Using the rule of thumb that the between-test correlation can be at most the product of the reliabilities of the two tests, would indicate that the pupil measure and total AQ score should have reliabilities of around. 8. This seems high for these kind of data. Is there evidence that this is so? Or is the correlation reported here an overestimate despite the substantial sample size (for which the authors should be applauded). This of course doesn't mean that the correlation isn't real, and even a substantially weaker correlation would be useful information here. The authors should review the correlations, and report reliability information if available to provide an estimate of what a realistic upper bound for the correlation is. It may be that the theoretical upper bound is lower than the observed correlation.

4) It is impressive that pupil size changes explains so much variance in autistic traits, not only because other behavioral measures did not seem to be as successful (as reviewed briefly in the Discussion), but also because pupil size changes are believed to be a noisy measure. Given that the authors "re-used" a subsample of their participants in the "swapped motion direction" control experiment, and yet another subsample in the "double task" control experiment, it would be nice to present the correlation of the test-retest sessions for each of these repeated experiments. We expect the pupil size changes to correlate very strongly between sessions, thus offering an upper-bound on the correlation one might expect to measure between pupils size changes and autistic traits.

5) One reviewer noted concerns about the availability of the data. *eLife* policy is that all data and software crucial to understand and replicate the findings of a manuscript ought to be publicly available (*eLife*'s data availability policy can be found here: https://submit.elifesciences.org/html/*eLife*_author_instructions.html#policies; please refer to sections "Availability of Data, Software, and Research Materials" and "Data Availability"). Please clarify whether there are any constraints preventing you from making this data publicly available. If not, the data should be provided in the form of supplementary files, source data files or source code (when applicable) with the submission or deposited to an external repository. A comprehensive catalogue of databases has been compiled by the BioSharing information resource (https://fairsharing.org/biodbcore/) but *eLife* Editorial staff can offer more specific guidance as needed.

[Editors' note: further revisions were requested prior to acceptance, as described below.]

Thank you for resubmitting your work entitled "Pupillometry reveals perceptual differences that are tightly linked to autistic traits in typical adults" for further consideration at *eLife*. Your revised article has been favorably evaluated by Richard Ivry as the Senior Editor, Nicole Rust as the Reviewing Editor, and three reviewers.

The manuscript has been improved but there are some remaining issues that need to be addressed before acceptance, as outlined below:

1) The description of the new experiment seems to be missing from the Materials and methods. It is not difficult to figure out the details of the experiment, but, for completeness, it should be in the Materials and methods.

2) Regarding the experimental materials: It's great that you have now made these available, including the experimental scripts. But the materials can still use a bit of editing to make them more accessible. Here are some ideas on how you might improve things (although you should feel free to do this however you think best):

- Add a README file that clearly explains the dependencies (i.e. what software do you need), how the analysis should be executed, and the file/ folder structure (i.e. what is located where).

- Add the raw data. Right now the data is provided in.mat format. But it would be better to also (or only) include the datafiles as they are created by the eye tracker (i.e. EDF for the EyeLink), and explain how these can be converted to.mat for further analysis.

- Add a LICENSE file to specify the license, such as CC-by.

---

## [Author Response]

[…] At the same time, the reviewers noted the following concerns that need to be addressed before publication:1) The key assumption of the paper is that switch-triggered pupil changes are indicative of individual differences in default perceptual styles. The authors support this by having additional experiments that bias all subjects toward the same perceptual style-which is a nice control. An additional strong support for this key assumption would be within-subject evidence that if a person switches her/his perceptual style that would result in a predicted modulation of switch-triggered pupil changes. Such a within-subject control would not need as many subjects, and would show that self-initiated changes in the perceptual style yield predictable pupil results-further supporting the main assumption of the paper. (This could be done with or without an objective confirmation that the perceptual style changed).

Thank you for this suggestion. We now have retested 10 of the original subjects in the way suggested. Despite the fact that all claimed they were unable to change their viewing style, there was a clear and significant change in the group results, providing additional support for the theory. Thank you.

2) The relative directions of pupil changes in Figure 1 and Figure 3 are consistent with the authors' assumptions, but the sign of those changes is not. That is, in Figure 1, shouldn't going from black in front to white in front result in a decrease in pupil size? It appears that we're seeing two effects, one that is modulated by dot color and a main effect of switch that causes an increase in pupil size. One can come up with a reasonable explanation for that, but it needs to be included in the manuscript. If there are indeed two effects there, it would be interesting to see if they can be analytically separated and correlated with AQ.

We are indeed seeing two main effects, and apologize for not making that clearer. One effect is a dilation on every perceptual switch, an effect first reported for binocular rivalry by Olivia Carter in 2008. Superimposed on this general dilation that occurs on every switch there is a luminance-specific modulation, the one we are interested in. The luminance-specific modulation (difference) is correlated with AQ (Figure 1), the general modulation on each switch (mean) is not (Figure 1). Perhaps this was lost because Figure 1 was too crowded. We have therefore moved Figure 1 to a separate figure, and attempted to bring out the fact that there are two effects more clearly.

3) There is concern with the strength of the correlation observed between the pupil measure and the total AQ score. The. 7 value seems extremely strong for this type of effect, and such a strong correlation between two measures is only possible if the measures themselves are highly reliable. Using the rule of thumb that the between-test correlation can be at most the product of the reliabilities of the two tests, would indicate that the pupil measure and total AQ score should have reliabilities of around. 8. This seems high for these kind of data. Is there evidence that this is so? Or is the correlation reported here an overestimate despite the substantial sample size (for which the authors should be applauded). This of course doesn't mean that the correlation isn't real, and even a substantially weaker correlation would be useful information here. The authors should review the correlations, and report reliability information if available to provide an estimate of what a realistic upper bound for the correlation is. It may be that the theoretical upper bound is lower than the observed correlation.4) It is impressive that pupil size changes explains so much variance in autistic traits, not only because other behavioral measures did not seem to be as successful (as reviewed briefly in the Discussion), but also because pupil size changes are believed to be a noisy measure. Given that the authors "re-used" a subsample of their participants in the "swapped motion direction" control experiment, and yet another subsample in the "double task" control experiment, it would be nice to present the correlation of the test-retest sessions for each of these repeated experiments. We expect the pupil size changes to correlate very strongly between sessions, thus offering an upper-bound on the correlation one might expect to measure between pupils size changes and autistic traits.

These are important points. We were also concerned that the correlation is almost “too good”.

We address these issues in a couple of ways. We did not attempt a comparison between results in the main and the double-task experiment, given that we propose that the latter induced a change of strategy (implicitly forcing a local style). Nor did we pursue a “split-half” approach for reliability, as this reduces the data size and makes the correlation more noisy. We did however correlate the results of the swapped motion experiment with those of the main experiment in the 22 participants who performed both, yielding a strong and highly significant correlation (r = 0.68 [0.36 0.86], p < 0.001, BF = 62.7). We agree that this should be expected to be higher than the correlation against AQ: however, given the confidence limits (now reported), the results are not unreasonable.

AQ scores have a strong internal-consistency and good test-retest reliability. For example: the British test-retest study (r=0.75; Baron-Cohen et al. 2001), the Japanese (r=0.77; Kurita et al. 2005) and the Dutch (r=0.78; Hoekstra et al. 2008).

Importantly, we now give 95% confidence intervals of all results, showing that even with this quite large sample we cannot be certain of the exact correlations. We are, however, certain that they are significantly positive.

5) One reviewer noted concerns about the availability of the data. eLife policy is that all data and software crucial to understand and replicate the findings of a manuscript ought to be publicly available (eLife's data availability policy can be found here: https://submit.elifesciences.org/html/eLife_author_instructions.html#policies; please refer to sections "Availability of Data, Software, and Research Materials" and "Data Availability"). Please clarify whether there are any constraints preventing you from making this data publicly available. If not, the data should be provided in the form of supplementary files, source data files or source code (when applicable) with the submission or deposited to an external repository. A comprehensive catalogue of databases has been compiled by the BioSharing information resource (https://fairsharing.org/biodbcore/) but eLife Editorial staff can offer more specific guidance as needed.

We agree with this policy, and the data is now all available online (through Dryad as suggested by the *eLife* policy). Here is the link: http://datadryad.org/review?doi=doi:10.5061/dryad.b3ng3. Now it is reported in the transparent reporting form.

[Editors' note: further revisions were requested prior to acceptance, as described below.]

The manuscript has been improved but there are some remaining issues that need to be addressed before acceptance, as outlined below:1) The description of the new experiment seems to be missing from the Materials and methods. It is not difficult to figure out the details of the experiment, but, for completeness, it should be in the Materials and methods.

Sorry for this omission: the description of this experiment has now been added to the Materials and methods section, sixth paragraph.

2) Regarding the experimental materials: It's great that you have now made these available, including the experimental scripts. But the materials can still use a bit of editing to make them more accessible. Here are some ideas on how you might improve things (although you should feel free to do this however you think best):- Add a README file that clearly explains the dependencies (i.e. what software do you need), how the analysis should be executed, and the file/ folder structure (i.e. what is located where).- Add the raw data. Right now the data is provided in.mat format. But it would be better to also (or only) include the datafiles as they are created by the eye tracker (i.e. EDF for the EyeLink), and explain how these can be converted to.mat for further analysis.- Add a LICENSE file to specify the license, such as CC-by.

Thank you, we appreciate your help in organizing this material. We have added a README file, indicating how files are organized, and a LICENSE file.

We never collected separate EDF files, but streamed Eyelink data to Matlab, storing the output for each trial as a Matlab matrix. The original. mat files contained personal information about the participants (initials and date of birth), which we are not allowed to disclose. We provide anonymized files, where the Eyelink data in the original trial-by-trial format is given in the “r” structure; in addition, we also provide a “ms” structure with pre-processed data: aligned to phase onset, which we used for our analyses. All this is now detailed in the README file.